# Molecular Basis of Methicillin and Vancomycin Resistance in *Staphylococcus aureus* from Cattle, Sheep Carcasses and Slaughterhouse Workers

**DOI:** 10.3390/antibiotics12020205

**Published:** 2023-01-18

**Authors:** Hanan A. Zaher, Shimaa El Baz, Abdulaziz S. Alothaim, Sulaiman A. Alsalamah, Mohammed Ibrahim Alghonaim, Abdullah S. Alawam, Mostafa M. Eraqi

**Affiliations:** 1Food Hygiene and Control Department, Faculty of Veterinary Medicine, Mansoura University, Mansoura 35516, Egypt; 2Department of Hygiene and Zoonoses, Faculty of Veterinary Medicine, Mansoura University, Mansoura 35516, Egypt; 3Department of Biology, College of Science in Zulfi, Majmaah University, Majmaah 11952, Saudi Arabia; 4Department of Biology, College of Science, Imam Mohammad Ibn Saud Islamic University, Riyadh 11623, Saudi Arabia; 5Microbiology and Immunology Department, Veterinary Research Institute, National Research Centre, Dokki, Giza 12622, Egypt

**Keywords:** *Staphylococcus aureus*, MRSA, VRSA, cattle, sheep

## Abstract

*Staphylococcus aureus* (*S. aureus*) is a serious infection-causing pathogen in humans and animal. In particular, methicillin-resistant *S. aureus* (MRSA) is considered one of the major life-threatening pathogens due to its rapid resistance to several antibiotics in clinical practice. MRSA strains have recently been isolated in a number of animals utilized in food production processes, and these species are thought to be the important sources of the spread of infection and disease in both humans and animals. The main objective of the current study was to assess the prevalence of drug-resistant *S. aureus*, particularly vancomycin-resistant *S. aureus* (VRSA) and MRSA, by molecular methods. To address this issue, a total of three hundred samples (200 meat samples from cattle and sheep carcasses (100 of each), 50 hand swabs, and 50 stool samples from abattoir workers) were obtained from slaughterhouses in Egypt provinces. In total, 19% *S. aureus* was isolated by standard culture techniques, and the antibiotic resistance was confirmed genotypically by amplification *nuc*A gen. Characteristic resistance genes were identified by PCR with incidence of 31.5%, 19.3%, 8.7%, and 7% for the *mec*A, *Van*A, *erm*A, and *tet* L genes, respectively, while the *aac6-aph* gene was not found in any of the isolates. In this study, the virulence genes responsible for *S. aureus’* resistance to antibiotics had the highest potential for infection or disease transmission to animal carcasses, slaughterhouse workers, and meat products.

## 1. Introduction

*Staphylococcus aureus* is one of the most common causes of foodborne disease worldwide [1]. This is crucial for the livestock and food industries in particular, as outbreaks could cause significant financial losses [2]. The *S. aureus* infection is considered to be one of the most important zoonotic diseases among farm animals and human accomplices. In animals, the bacterium causes mastitis, sepsis, followed by wound infection [3]. Via the food chain, animal to human infection may lead to food poisoning, bacteremia, pneumonia, arthritis, sepsis, and nosocomial infections [4].

Furthermore, *S. aureus* infection may affect the productivity and quality of the meat, restricting international supply [5,6]. Consumers and food handlers may become easy targets for the spread of *S. aureus* through the food chain [7]. It has been proven that *Staphylococcus* contamination either during or after preparation is an indication of poor food hygiene standards, making the meat unfit for refrigeration and transportation [8]. 

More than half of *S. aureus* isolates found in animal-origin foods, especially raw meat products, are enterotoxigenic [9]. *S. aureus* produces enterotoxins (SEs), which are harmful to both human and animal digestive systems [10]. 

The protection of public health and the safety of food are significantly impacted by the rise in multidrug resistance (MDR) bacterial strains in farm animals [11]. The abuse of antibiotics in livestock, either for growth promotion or treatment, may be lead to increased *S. aureus* resistance to antibiotics [12].

The first discovered antibiotic was penicillin, and in the year 1940 *S. aureus* emerged as a clinical issue as a result of developing resistance to penicillin, by creating β-lactamase enzyme that rendered penicillin inactive [13,14,15]. Additionally, the prevalence of MRSA strains that are resistant to a variety of non-β-lactam antibiotics has continuously increased [13]. *S. aureus* has a wide range of different virulence factors, including staphylococcal protein A (*spa*), in addition to its ability to resist a variety of antibiotics [16]. *S. aureus* from animal sources carries resistance genes, such as methicillin-resistance (*mecA*), vancomycin-resistance (*van*), penicillin-resistance (*blaZ*), chloramphenicol/florfenicol-resistance (*cfr*), apramycin-resistance (*apm*), and spectinomycin-resistance (*spc*) genes [17,18,19,20,21,22]. In addition, *S. aureus* exhibits a variety of virulence genes, including SEs, Panton-Valentine leucocidin (*pvl*), clumping factor A (*clfA*), and toxic shock syndrome toxin (*tst*) [23,24,25]. 

Methicillin resistance is mediated by the *mec*A gene [26], which was acquired through horizontal transfer of the staphylococcal cassette chromosome *mec* (SCC*mec*) [27]. β-lactams antibiotics have a poor affinity for the alternative penicillin-binding protein 2a (PBP 2A) [28] that the *mec*A gene produces, leading to antibiotic resistance in the entire family [29]. MRSA strains are frequently resistant to antibiotics other than β-lactams, such as oxazolidinones, lipopeptides, glycopeptides lincosamides, aminoglycosides, and macrolides [30,31].

In recent years, the frequency of *S. aureus* mutants resistant to vancomycin has progressively increased [32,33]. The antagonistic effects of *mec*A and *van*A resistance determinants may be the cause of VRSA infections in clinical settings [15,34]. 

In *S. aureus*, the aminoglycoside-modifying enzyme (AME) often mediates resistance to aminoglycosides [35]. The first instance of gentamicin-resistant *S. aureus* was noted in 1975 [36]. The *aac* 6-*aph* 2 gene is the most prevalent in aminoglycoside-resistant *S. aureus* isolates [37,38]. Multiple processes can lead to macrolide resistance, such as erythromycin, but the most common one is target alteration produced by one or more *erm* genes that encode a 23S rRNA methylase, and this resistance type is referred to as MLSB resistance phenotypically [39]. It has also been proven that staphylococci with macrolide efflux pumps (encoded by *msr*A or *msr*B) exhibit resistance to MLS antibiotics [40]. 

Tetracycline resistance genes are present in the majority of bacteria with tetracycline resistance genes (*tet*) [41,42]. 

Major public health problems concern food-associated microorganisms that carry genes for antibiotic resistance. This is due to the fact that they can disseminate antibiotic resistance genes to commensal and enteric bacteria through horizontal gene transfer of mobile genetic elements and can result in foodborne diseases [43]. The frequency of MDR changes over time and varies based on the host species and place of origin, due to variances in antimicrobial consumption and antibiotic abuse [44]. The cost of treating these conditions is significant. As a result, it is critical to keep track of the prevalence and antibiotic resistance of food-borne pathogens in order to plan effectively for food safety and to implement focused interventions [45]. Furthermore, there are no data available in Egypt on the prevalence of MRSA and VRSA at the level of the abattoir.

The main objective of the current study was to assess the prevalence of drug-resistant *S. aureus*, particularly vancomycin-resistant *S. aureus* (VRSA) and MRSA, by molecular methods from cattle, sheep carcasses, and slaughterhouses workers.

## 2. Results

### 2.1. Prevalence of S. aureus in Examined Sample

A total of 300 samples (200 meat samples from cattle and sheep carcasses (100 of each), 50 hand swabs, and 50 stool samples from abattoir workers) were obtained from Egypt governorates and screened for *S. aureus*. Among these, 57 (19%) *S. aureus* strains were identified using morphological, biochemical, and molecular methods, including *nuc*A gene detection. The findings demonstrated that all strains carried the *nuc*A gene. *S. aureus* was present in all of the examined samples with a prevalence of 20% (20/100), 15% (15/100), 34% (17/50), and 10% (5/50) from examined cattle carcass, sheep carcass, hand swabs, and stool samples, respectively (Table 1).

### 2.2. Molecular Characterization of Isolated S. aureus

All examined isolates were owed the *nuc*A gene. Of the detected *S. aureus*, 35% (7/20), 26.6 (4/15), 29.4 (5/17), and 40% (2/5) were carried to the *mec*A gene in cattle, sheep carcass, hand swab, and human stool, respectively. *van*A were determined in 35% (7/20), 20% (3/15), 11.7% (2/17) of cattle, sheep carcass, and hand swabs, respectively, while none of the examined human stool samples were carried to *van*A gene. The percentages of isolates that owed *tet* (L) in cattle, sheep carcass, hand swabs, and human stool were 5% (1/20), 6% (1/15), 5.8% (1/17), and 20% (1/5), respectively; however, the *erm*A gene was found in 10% (2/20), 13.3% (2/15), and 5.8% (1/17), respectively, in same tested strains while it was not found in human stool, whereas the *aac6-aph* gene was not detected in any of the tested isolates (Table 1; Figure 1).

### 2.3. Antibiotic Susceptibility Test of S. aureus

For the antimicrobial susceptibility test, the tested isolates showed high resistance to cephalexin (78.9%) while they showed high susceptibility to telavancin and daptomycin (100% per each) (Table 2). The MAR index values showed multiple resistant patterns, revealing that the MAR index average of *S. aureus* was 0.343 (Table 3).

## 3. Discussion

Cutler et al. (2010) [46] estimated that about 60% of emerging human diseases are zoonotic. The use of antibiotics for treatment and as growth promoters in livestock may encourage the spread of novel diseases in animals, particularly those that are resistant to antibiotics [47]. The current phenotypic study found that the prevalence of *S. aureus* was 19% among all samples assessed. The genotypic amplification proved to have 100% of the *nucA* gene in all the isolates of *S. aureus*. In contrast to earlier findings, *S. aureus’* prevalence in raw meat was 15% in Egypt [48] and 20.5% in China [49]. However, other studies reported a prevalence comparable to the present research finding; for example, a higher prevalence of *S. aureus* found in raw red meat was 29.4% in Algeria [50], 32.8% in Japan [51], 34.3% in Ethiopia [52], 40.38% in Morocco [53], 45% in Ghana [54], and a low prevalence was 1.3% in Nigeria [55]. 

In the current study, among 57 *S. aureus* isolates, only 18 isolates (31.5%) have the *mecA* gene. MRSA prevalence based on *mec*A identification is the highest in human stools (40%), followed by beef (35%), hand swab (29.4%), and sheep (26.6%). According to a similar study conducted in African countries, beef has a higher potential risk of transmitting MRSA to humans than other red meats, with a total prevalence of 33.08% of MRSA in beef samples and 24.5% in other red meat samples [56]. Previous studies have reported that the highest prevalence for the *mec*A gene was detected amongst the isolates of cows, buffaloes, and humans (80% each). However, just one isolate (20%) was recovered from a sheep. In a study carried out in Pakistan, the prevalence of MRSA was 63% in beef, 50% in mutton, 45% in knives, 28% in the working area, 25% in rectal swabs, 18% in hooks, and 18% in butchers’ hands [57]. Similarly, earlier research also discovered 33.3% MRSA isolates from hand swabs [58]. 

In the current study, among 57 *S. aureus* isolates, only 11 isolates (19.3%) have the *van*A gene, the highest percentage (35%) was found in cattle samples, followed by sheep (20%), and hand swab (11.7%). However, the gene was not detected in human stool samples. Unlike our findings, higher recovery rates (60%) and (40%) were reported in hand swabs of food handlers and diarrheic stool in Egypt, respectively [59]. In previous studies, the highest prevalence was noticed among *S. aureus* isolated from sheep (100%), while isolates recovered from cow, buffalo, and goat milk revealed a prevalence rate of 40% for each host species. On the opposite side, all the tested human samples were negative for the *van*A gene [60,61]. In another study carried out by J. Kaszanyitzky et al. (2007), the prevalence of *S. aureus* was 14.5% in camel meat and 55% in slaughterhouse workers. The VRSA incidence was 27% and 54% in camel meat and slaughterhouse workers, respectively, which were identified as MRSA with presence of the *van*A gene. The analysis of the *van*A gene sequences from camel meat and humans revealed that they were identical to each other, suggesting the zoonotic importance of this pathogen and/or horizontal gene transfer [62]. In general, VRSA in livestock can spread through the hands of slaughterhouse workers or through the consumption of viscera-contaminated meat during the process. The bacterial colonization could enhance the risk of zoonotic disease transmission [63].

An MAR index above 0.2 is a parameter used to reveal the spread of bacterial resistance in certain populations, and this bacteria originated from a habitat in which different antibiotics are used and often abused [64]. A high incidence of MAR is permitted by genetic exchange between MAR pathogens and other bacteria [65]. In this study, the MAR indices of *S. aureus* isolates were 0.343, revealing that these isolates were derived from samples from high-risk sources. These findings are consistent with a past study, which found the MAR index of the majority of the *S. aureus* isolated from pork meat was 0.3 [66].

The microbial contamination at the slaughter, storage, or at the meat processing units are potential sources of meat contamination in and around Africa and throughout the world [67]. As per our understanding, the antimicrobial drugs are irrationally utilized internationally to prevent bacterial and fungal contamination in meat and meat products, which may further lead to the high incidence of MRSA and VRSA prevalence [10,12]. It is imperative to note some limitations of this study. In this research, the phenotypic and molecular analyses were carried out in vitro to prove the virulence efficiency, and not in vivo, due to the research objective, facilities, and limited funding. While we identified some important antimicrobial resistance genes and a limited number of *S. aureus*, a total of 300 samples were analyzed for this research with informed consent and agreement from human participants and slaughter market proprietors. In the future, these limitations can be controlled by using a larger number of *S. aureus* isolates from different demographic regions in Egypt provinces to predict the antimicrobial resistance and virulence trend. Furthermore, it would allow for further genotyping studies of multidrug resistance strains using multilocus sequence typing and whole-genome sequencing methods, to offer data for better understanding and evaluating global methicillin-resistant *S. aureus*/methicillin-sensitive *S. aureus* strains. Moreover, guidelines and programs that prevent the abuse of antibiotics to prevent the occurrence of antimicrobial resistance in livestock must be developed.

## 4. Materials and Methods

### 4.1. Samples Collection and Preparation

#### 4.1.1. Animal Samples

For the bacteriological analysis, the sheep and cattle samples were collected, packed in sterile bags, and transported at 4 °C using an ice box to the meat hygiene laboratory, Food Hygiene and Control Department, College of Veterinary Medicine, Mansoura University, Egypt.

#### 4.1.2. Human Samples

From 50 adult male slaughterhouse workers, hand swabs were taken. Both hands’ palm surfaces were cleaned using cotton-tipped swabs that had been saturated with TSB. For sampling, sterile gloves were worn to reduce sample contamination. All stool samples were transported on ice to the laboratory for additional analysis after being collected in sterile cups (sample collection was carried out after handling meat for at least an hour and all workers were clinically free of any bacterial skin infections at the time of examination and asked not to wash their hands before sampling). 

### 4.2. Isolation and Identification of S. aureus

Each sample was put into 9 mL of Tryptone Soya Broth (TSB) with 70 mg of NaCl per mL, and it was all left to sit for 24 h at 37 °C. After incubation, a loopful from all inoculated samples were cultured on selective media for *S. aureus*, Baird-Parker agar (oxoid, 277) supplemented with 5% egg yolk-tellurite emulsion and incubated at 37 °C for 24–48 h. Black, shiny, cylindrical, smooth, convex colonies with a zone of clearing around them were selected and purified for additional biochemical analysis. The biochemical tests, such as catalase activity, deoxyribonucleases (DNase), coagulase production, thermostable nuclease production, and the mannitol fermentation test, were performed for phenotypic identification [68].

### 4.3. Molecular Characterization of Isolated S. aureus:

The phenotypically identified *S. aureus* isolates were examined by PCR for the *nuc*A gene then all positive isolates for the *nuc*A gene were further examined for other virulence and resistance genes.

#### 4.3.1. DNA Extraction

Bacterial DNA was prepared by method mentioned before by Guran and Kahya [69], in which three to five colonies of biochemically identified *S. aureus* isolates were mixed with 100 μL of sterilized distilled water, then exposed to heat block at 95 °C for 15 min. After that, heated lysates were centrifuged at 13,000 rpm for 10 min. The supernatants were transferred to clean tubes and kept at −20 °C for use as DNA template.

#### 4.3.2. DNA Amplification

PCR was carried out according to the method mentioned before by Guran and Kahya with the primers listed in Table 1 and performed in an ABI Veriti Thermal Cycler (Applied Bio-systems Asia Pte Ltd., Singapore). A total reaction volume of 50 µL consisting of 6 µL of 10 × PCR buffer, 8 µL of 25 mmol/L MgCl2, 8 µL of 10 mmol/L deoxynucleoside triphosphate mixtures, 1 µL Taq polymerase, 5 µL of template DNA (5 ng/µL), 1 µL of 25 pmol each gene, and 12 µL of molecular grade water was used in the PCR. PCR reaction was started with an initial denaturation at 94 °C for 5 min, followed by 30 cycles of denaturation at 94 °C for 45 s, annealing at 54 °C for 45 s, 62 °C for 45 s, extension at 72 °C for 45 s, and a final extension at 72 °C for 7 min. PCR products were separated and visualized in a 1.5% agarose gel (Table 4).

### 4.4. Reference Strain

For the present antimicrobial susceptibility tests (AST), the following reference strains were used: 1. *Staphylococcus aureus*-ATCC: 25,923 for MSSA, 2. ATCC: 43,300 for MRSA. The reference strains were obtained from the Food Analysis Center, Faculty of Veterinary Medicine, Benha University, Egypt.

### 4.5. Antimicrobial Susceptibility Testing

All positive isolates for *S. aureus* (n = 57) were tested for antimicrobial susceptibility by the agar disk diffusion method using disks as shown in Table 5. The procedure followed guidelines from the CLSI Inst [70]. Multiple Antibiotic Resistance (MAR) index for each strain was determined according to the following formula: MAR index = No. of resistance/Total No. of antibiotic [71,72].

## 5. Conclusions

*Staphylococcus aureus* in meat, especially MRSA and VRSA, poses a zoonotic and public health threat and raises concerns about food safety. It is directly linked to poor hygienic practices by individuals involved in various stages of meat processing, such as production, handling, shipping, slicing, storage, and point of sale. To prevent food contamination with *S. aureus*, continual tracking and deployment of better management techniques inside the food chain are necessary. The cutting edge molecular-based diagnostics is the need of the hour in order to deliver high quality meat products to ensure the public health and food safety of end users. 

## Figures and Tables

**Figure 1 antibiotics-12-00205-f001:**
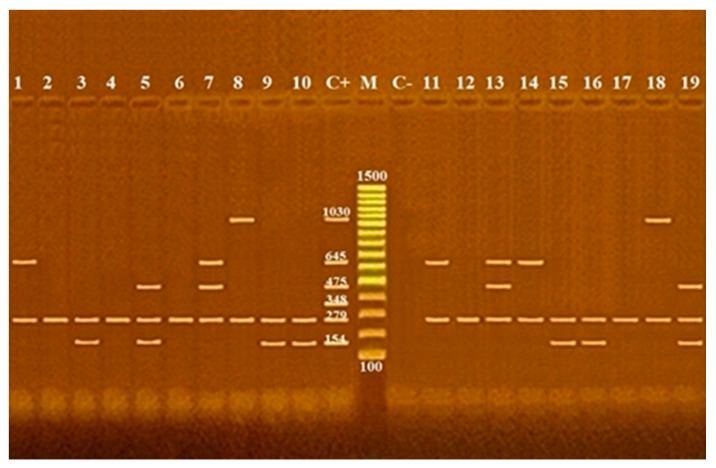
Agarose gel electrophoresis of multiplex PCR of *nuc*A (279), *mec*A (154 bp), *aac 6-aph* 2 (348 bp), *tet* L (475 bp), *erm*A (645 bp), and *van*A (1030 bp) antibiotic resistance genes of *S. aureus*. Lane M: 100 bp ladder as molecular size DNA marker. Lane C+: Control positive for *nuc*A, *mec*A, *aac 6-aph* 2, *tet* L, *erm*A, and *van*A genes. Lane C−: Control negative. Lanes from 1 to 19: positive *S. aureus* strains for *nuc*A gene. Lanes 2, 5, 9, 10, 15, 16, and 19: positive *S. aureus* strains for *mec*A gene. Lanes 5, 7, 13, and 19: positive *S. aureus* strains for *tet* L gene. Lanes 1, 7, 11, 13, and 14: positive *S. aureus* strains for *erm*A gene. Lanes 8 and 18: positive *S. aureus* strains for *van*A gene. Lanes from 1 to 19: negative *S. aureus* strains for *aac 6-aph* 2 gene.

**Table 1 antibiotics-12-00205-t001:** Prevalence of *S. aureus* and its antibiotic resistance genes isolated from cattle, sheep carcass, human stools, and hand swabs samples.

Types of Samples	No. of Examined Samples	*S. aureus* Isolates	*nuc*A	*mec*A	*Tet* (L)	*erm*A	*van*A	*aac 6-aph*
		%	%	%	%	%	%	%
Cattle	100	20	100	35	5	10	35	0
Sheep	100	15	100	26.6	6	13.3	20	0
Hand swabs	50	34	100	29.4	5.8	5.8	11.7	0
Human stools	50	10	100	40	20	0	0	0
Total	300	19	100	31.5	7	8.7	19.3	0

**Table 2 antibiotics-12-00205-t002:** Antibiogram susceptibility of *S. aureus* strains (n = 57).

Antimicrobial Agent	Symbol	S	R
		%	%
Cephalexin	CE	15.8	78.9
Oxacillin	OX	26.3	73.7
Cephalothin	CN	36.8	52.6
Flucloxacillin	FL	47.4	47.4
Sulphamethoxazol	SXT	47.4	36.8
Dicloxacillin	DC	52.6	36.8
Erythromycin	E	63.2	31.6
Clindamycin	CL	63.2	26.3
Lincomycin	L	68.4	26.3
Cefazolin	CZ	73.7	21.1
Ceftibiprole	CB	84.2	10.5
Dalbavancin	D	84.2	5.3
Tigecycline	TG	89.5	5.3
Oritavancin	O	94.7	5.3
Linezolid	LZ	89.5	-
Telavancin	T	100	-
Deptomycin	D	100	-

S: sensitive. R: resistance.

**Table 3 antibiotics-12-00205-t003:** MAR index of *S. aureus* isolates (n = 57).

No. of Strain	Antimicrobial Resistance Profile	MAR Index
6	CE, OX, CN, FL, SXT, DC, E, CL, L, CZ, CB, D, TG, O	0.823
3	CE, OX, CN, FL, SXT, DC, E, CL, L, CZ, CB	0.647
5	CE, OX, CN, FL, SXT, DC, E, CL, L, CZ	0.588
4	CE, OX, CN, FL, SXT, DC, E, CL, L, CZ	0.588
4	CE, OX, CN, FL, SXT, DC, E, CL, L	0.474
3	CE, OX, CN, FL, SXT, DC, E	0.412
3	CE, OX, CN, FL, SXT, DC	0.353
4	CE, OX, CN, FL, SXT	0.294
4	CE, OX, CN, FL, SXT	0.294
4	CE, OX, CN	0.158
3	CE, OX	0.117
4	CE, OX	0.117
3	CE, OX	0.117
3	CE, OX	0.117
4	CE	0.059
Average		0.343

**Table 4 antibiotics-12-00205-t004:** Oligonucleotide primers used for identification for *S. aureus*.

Primer	Oligonucleotide Sequence (5′ → 3′)	Product Size (bp)	References
*nucA*	(F): 5′ GCGATTGATGGTGATACGGTT ′3	279	[69]
	(R): 5′ AGCCAAGCCTTGACGAACTAAAGC ′3		
*mecA*	(F): 5′ TAGAAATGACTGAACGTCCG ′3	154	
	(R): 5′ TTGCGATCAATGTTACCGTAG ′3		
*aac 6-aph 2*	(F): 5′ CAGAGCCTTGGGAAGATGAAG ′3	348	
	(R): 5′ CCTCGTGTAATTCATGTTCTGGC ′3		
*Tet* (L)	(F): 5′ CATTTGGTCTTATTGGATCG ′3	475	
	(R): 5′ ATTACACTTCCGATTTCGG ′3		
*ermA*	(F): 5′ TCTAAAAAGCATGTAAAAGAA ′3	645	
	(R): 5′ CTTCGATAGTTTATTAATATTAGT ′3		
*vanA*	(F): 5′CATGAATAGAATAAAAGTTGCAATA′3	1030	
	(R): 5′ CCCCTTTAACGCTAATACGATCAA ′3		

**Table 5 antibiotics-12-00205-t005:** Antimicrobial discs, concentration, and interpretation of their action on the isolated pathogens (μg).

Antibiotics	Symbol	Concentration
Tigecycline	TG	30
Lincomycin	L	15
Cefazolin	CZ	30
Ceftibiprole	CB	30
Dicloxacillin	DC	1
Linezolid	LZ	30
Telavancin	T	5
Cephalothin	CN	30
Flucloxacillin	FL	1
Clindamycin	CL	10
Dalbavancin	D	30
Oxacillin	OX	1
Oritavancin	O	5
Deptomycin	D	30
Erythromycin	E	15
Cephalexin	CE	30
Sulphamethoxazol	SXT	25

## Data Availability

Not applicable.

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
