# Peer review of "Molecular Basis of Methicillin and Vancomycin Resistance in Staphylococcus aureus from Cattle, Sheep Carcasses and Slaughterhouse Workers"

_antibiotics, 2023, doi:10.3390/antibiotics12020205_

Round 1

Reviewer 1 Report

Authors have presented an important work that have potential impact on maintaining safe practices in livestock industries. 

However, Authors whould be careful about the language, there are a lot of grammatical mistakes/typos in the manuscript, for example:

Line 117, the main objective of this study was to use……

Line 124, …Provinces of Egypt?

In Table 1 and 2, column for both Number and percentage are redundant, mention of one should be sufficient.

Authors should comment on why vanA genes are found only in S. aureus isolated from Sheep but not from human.

i

Author Response

Dear, Reviewer  

Thank you for giving me the opportunity to submit a revised draft of my manuscript titled [Molecular basis of methicillin and vancomycin Staphylococcus aureus resistance in cattle, sheep carcass and slaughterhouse workers] to [Antibiotics-MDPI]. We appreciate the time and effort that you have dedicated to providing your valuable feedback on my manuscript. We are grateful to you for their insightful comments on us. We have been able to incorporate changes to reflect most of the suggestions provided by you. We have highlighted the changes within the manuscript.

Sincerely,

Dr. Mostafa M. Eraqi

Reviewer 2 Report

Some annotations are made on the document, or they are marked (in color), so that the authors review and correct them in the best way, since there are some details of writing, style, syntax and grammar.

It is important to review the order of the sections, since it appears:

Introduction, results, discussion, material and methods, conclusions, and references; when in strict order it should be: Introduction, Material and methods, Results, Discussion, Conclusions, and References.

It is important to consider the guidelines of writing - editing that gives the journal, and as an example the following conditions can be applied at work: This  predominance  of A. lumbricoides than  any  other  GIPs  agreed  with some  other  reports  [15,22].  On  the  other  hand, da   Silva   et   al.   [19],   Khanal   et   al.   [2]   and Hernández    et    al.    [25]    found T.    trichiura predominant

Author Response

Dear, Reviewer  

Thank you for giving me the opportunity to submit a revised draft of my manuscript titled [Molecular basis of methicillin and vancomycin Staphylococcus aureus resistance in cattle, sheep carcass and slaughterhouse workers] to [Antibiotics-MDPI]. We appreciate the time and effort that you have dedicated to providing your valuable feedback on my manuscript. We are grateful to you for your insightful comments on us. We have been able to incorporate changes to reflect most of the suggestions provided by you. We have highlighted the changes within the manuscript.

Sincerely,

Dr. Mostafa M. Eraqi

Reviewer 3 Report

Zaher et al. have done a nice piece of work and it is a current need to do continuous surveillance programs to understand the antimicrobial resistance and dissemination in one health perspective. However, the current study needs some minor revisions before the acceptance of the paper.

Title

1.     Please change the title of the paper to “Molecular basis of methicillin and vancomycin resistance in Staphylococcus aureus in cattle, sheep carcasses and slaughterhouses workers”

Abstract

1.     Line 17: Bacterial scientific names should be italicized

2.     Please modify the abstract in a way that has explain your aims, objectives and outcomes.

Introduction

1.     Line 46: What is MDR? Please expand and give a proper definition.

2.     Line 46: Please change to misuse and over use of antibiotics instead of the word “abuse”.

3.     Introduction is too lengthy and advise to reduce the content and highlight the research gap, hypothesis, aims and objectives of the current study.

Results

1.     Please add a map of sample collection points and their prevalence according to the location for better understanding

2.     Line 126: Isn’t it the Table 1?

3.     Line 128: It should be nucA gene, right?

4.     Add either table 2 or figure 2 as it duplicates the results.

5.     If you are using figure two, what were your assumptions on intermediate susceptibility strains in table 2? Will those strains belong to resistant? In such cases, I recommend to categories susceptible and non-susceptible groups as you can assign intermediate strains in to non-susceptible group. However, the authors have to define the criteria.

Discussion

1.     Please add a take home massage of the study at the end of the discussion.

2.     What are the limitations of the current study.

3.     explain the translation of data you have generated in the current study to AMR mitigation policy development and the possible future research directions as the whole purpose of the study needs to address

4.     Please improve the discussion. Lengthy discussions are not always a good discussion. Limit the word count and highlight the important findings.

Methods

1.     Please mention the reference genomes used in the study

Author Response

Thank you for giving me the opportunity to submit a revised draft of my manuscript titled [Molecular basis of methicillin and vancomycin Staphylococcus aureus resistance in cattle, sheep carcass and slaughterhouse workers] to [Antibiotics-MDPI]. We appreciate the time and effort that you have dedicated to providing your valuable feedback on my manuscript. We are grateful to you for your insightful comments on us. We have been able to incorporate changes to reflect most of the suggestions provided by you. We have highlighted the changes within the manuscript.
